# Atomic Structure Evaluation of Solution-Processed *a*-IZO Films and Electrical Behavior of *a*-IZO TFTs

**DOI:** 10.3390/ma15103416

**Published:** 2022-05-10

**Authors:** Dongwook Kim, Hyeonju Lee, Bokyung Kim, Xue Zhang, Jin-Hyuk Bae, Jong-Sun Choi, Sungkeun Baang

**Affiliations:** 1Department of Electronic Engineering, Hallym University, Chuncheon 24252, Korea; d.kim@hallym.ac.kr (D.K.); hyeonjulee@hallym.ac.kr (H.L.); d21012@hallym.ac.kr (B.K.); 2College of Ocean Science and Engineering, Shangdong University of Science and Technology, Qingdao 266590, China; skd996027@sdust.edu.cn; 3School of Electronics Engineering, Kyungpook National University, Daegu 41566, Korea; jhbae@ee.knu.ac.kr; 4School of Electronic and Electrical Engineering, Kyungpook National University, Daegu 41566, Korea; 5Department of Electrical and Electronical Engineering, Hongik University, Seoul 04066, Korea

**Keywords:** solution-processed *a*-IZO films, thin-film transistor, metal–nitrate precursor

## Abstract

Understanding the chemical reaction pathway of the metal–salt precursor is essential for modifying the properties of solution-processed metal-oxide thin films and further improving their electrical performance. In this study, we focused on the structural growth of solution-processed amorphous indium-zinc-oxide (*a*-IZO) films and the electrical behavior of *a*-IZO thin-film transistors (TFT). To this end, solution-processed *a*-IZO films were prepared with respect to the Zn molar ratio, and their structural characteristics were analyzed. For the structural characteristic analysis of the *a*-IZO film, the cross-section, morphology, crystallinity, and atomic composition characteristics were used as the measurement results. Furthermore, the chemical reaction pathway of the nitrate precursor-based IZO solution was evaluated for the growth process of the *a*-IZO film structure. These interpretations of the growth process and chemical reaction pathway of the *a*-IZO film were assumed to be due to the thermal decomposition of the IZO solution and the structural rearrangement after annealing. Finally, based on the structural/chemical results, the electrical performance of the fabricated *a*-IZO TFT depending on the Zn concentration was evaluated, and the electrical behavior was discussed in relation to the structural characteristics.

## 1. Introduction

Recently, as the demand for flexible displays has increased and direct printing of display products has been attempted, interest in solution processes has drastically expanded [1,2,3,4]. In display manufacturing, using the solution process, time and cost can be reduced by more than 60% by direct printing at room temperature instead of using a vacuum deposition process [5]. In this regard, the possibility of solution process of amorphous oxide semiconductors (AOS) material has been paid much attention in displays, and various chemical processing methods have been developed for the solution process [6,7,8]. Because the solution process mostly targets flexible displays, lowering the process temperature is a major concern. However, as the production of the AOS film requires a process temperature of 500 °C or higher for the oxidation reaction, various attempts have been made to lower the process temperature by developing process techniques and chemical treatments [9,10]. In particular, developments of the chemical treatment such as metal–salt precursors, catalysts, and chemical chamber deposition contributed drastically to lowering the process temperature [11,12,13,14].

Among the many metal–salt reactions, the metal–nitrate precursor has many advantages over other conventionally used precursor reactions such as metal hydroxide, metal chloride, and metal acetate [15,16,17]. The metal–nitrate precursor has a high-density film and contains fewer impurities because of the high volatility of the reaction by-products. Moreover, the application of process developments allows the manufacture of high-quality films [18,19]. Therefore, for the solution-processed AOS TFTs, the nitrate precursors have been implemented as a manufacturing method for semiconductor films, dielectrics, and transparent electrodes. Recently, high-performance AOS TFTs fabricated at temperatures below 150 °C have been reported by applying improved processing techniques [20,21,22,23].

Eventually, the electrical properties of AOS TFTs are significantly affected by the bonding structure and composition ratio of the atoms constituting the semiconductor material [24,25]. AOS films manufactured by vacuum deposition form a relatively uniform amorphous random-network structure, whereas solution-processed AOS films are accompanied by many impurities/defects because the metal oxide is chemically rearranged while the solvent/by-product is removed [26]. Thus, the bonding structure/composition of the AOS film may vary depending on the annealing conditions, including the precursors, solvents, and catalysts used. Although understanding the chemical reaction pathway of the metal–nitrate reaction is the most effective way to fabricate an electrically enhanced film, the interpretation of the nitrate-precursor reaction pathway for a metal-oxide film is still controversial. There are arguments which exist depending on the by-product and chemical reactions [27,28]. Therefore, to fabricate an AOS film with improved performance, it is necessary to analyze the structural and electrical properties of the AOS film based on an accurate understanding of the metal–nitrate reaction pathway.

In this study, the electrical characteristics of amorphous *a*-IZO TFTs were analyzed based on the structural/stoichiometric characteristics of solution-processed *a*-IZO semiconductor films. For this, *a*-IZO films were prepared with respect to the Zn molarity, and structural characteristics such as the bonding structure, crystallinity, and composition were analyzed. Furthermore, based on the thickness-dependent properties and thermal decomposition results of the *a*-IZO film, it is speculated that the chemical reaction pathway and growth process of the film are determined. Finally, an *a*-IZO TFT with respect to Zn concentration was fabricated, and its electrical performance was evaluated. The electrical behavior of *a*-IZO TFT is discussed in relation to the structural/chemical characteristics of the *a*-IZO film. 

## 2. Materials and Methods

As shown in Figure 1a, the IZO solutions with respect to the Zn molarity ratio were prepared by dissolving indium-nitrate-hydrate (In_3_(NO_3_)_3_∙xH_2_O) (Sigma-Aldrich, St. Louis, MO, USA) and zinc nitrate hydrate (Zn_2_(NO_3_)_2_∙xH_2_O) (Sigma-Aldrich, St. Louis, MO, USA) in 2-methoxyethanol (CH_3_O(CH_2_)_2_OH) (Sigma-Aldrich, St. Louis, MO, USA). For proper dissolution, the solution was mixed using a magnetic bar at 60 °C for 12 h or longer. The molarity ratios, as listed in Table 1, the 0.05 M of the In molarity were used for characterization. Moreover, the range of the Zn molarity was adjusted so that the largest electrical change in the TFT could be observed. Additionally, the Zn molarity conditions of 0.1 M and 0.15 M were excluded because spin coating was not properly carried out in the experiments.

To fabricate the solution-processed *a*-IZO TFT, a *p*-type (p-doped) silicon (STC, Kyowa, Saku, Japan) wafer sputtered with a 100 nm thick SiN_x_ dielectric was used. Before spin coating the solution-processed semiconductor layer, the substrate was cleaned using an ultrasonicator (Hwashin Tech., Gwangju, Gyeonggi, Korea) in the order of acetone, isopropyl alcohol, and deionized water.

Thereafter, to lower the surface energy of the substrate, it was exposed to oxygen plasma (Daeki, Daejeon, Korea) at a chamber pressure of 3 mTorr for 1 min. Spin coating was carried out at 2000 rpm for 1 min to prepare the IZO solution, and soft banking was started for 5 min at 110 °C, immediately after coating. To anneal the semiconductor film, the spin-coated substrate was thermally annealed at 600 °C for approximately 1 h under a nitrogen atmosphere. Substrate heating and cooling were continued slowly for 1 and 6 h, respectively, and additionally cooled at room temperature for 6 h or more.

The fabricated source/drain of *a*-IZO TFT was a finger-type, as shown in Figure 1b, and aluminum was used for the electrodes. This aluminum electrode was processed by vapor-phase thermal deposition through a shadow mask at 1 × 10^−6^ Torr, and the thickness was approximately 50 nm. The width and length of the finger-type channel were 2000 and 80 μm, respectively, and W/L = 40.

The following equipment was used to analyze the fabricated *a*-IZO films. The surface and cross-sectional characteristics of the *a*-IZO films were measured using field-emission scanning electron microscopy (FE-SEM) (JS-6701F) (JEOL, Tokyo, Japan). Furthermore, the crystal structure and atomic composition of the prepared *a*-IZO films were analyzed using high-resolution X-ray diffraction (HR-XRD) (SmartLab, Tokyo, Japan) and X-ray photoelectron spectroscopy (XPS) (K-alpha) (ThermoFisher, Seoul, Korea), respectively. Auger electron spectroscopy (AES) (PHI 700) (Ulvac-PHI, MN, USA) was used to analyze the constituent atoms according to the depth of the *a*-IZO film, and the thermal decomposition process of the solution was analyzed using thermogravimetric analysis (TGA) (TGA N-1000) (Scinco, Seoul, Korea). Finally, all the electrical properties of the solution-processed *a*-IZO TFT were measured using a semiconductor analyzer (ELECS-420) (ELECS, Seoul, Korea) at room temperature.

## 3. Results and Discussion

### 3.1. Structural Properties

To discuss the structural characteristics of the *a*-IZO film, the properties of the fabricated film with respect to the Zn molarity were evaluated. Figure 2 shows the cross-sectional and surface SEM images of the *a*-IZO films with various Zn molar ratios. For this analysis, 0.2, 0.25, 0.3, 0.4, 0.5, and 0.6 M of the Zn molarity were used, and the In molarity ratio was fixed at 0.05 M. As shown in Figure 2a, the thickness of the *a*-IZO film increased from 20 to 70 nm as the Zn molar ratio increased from 0.2 to 0.6 M. The surface properties also exhibited noticeable differences depending on the Zn concentration, as shown in Figure 2b. Most of the dark part of 0.2 M was an *a*-IZO layer formed uniformly in the 20 nm thin film. This was confirmed by comparing the enlarged surface SEM images of the SiN_x_ and *a*-IZO. Even at a low Zn molarity ratio, the ZnO aggregates began to appear as bright seeds, and as the Zn molarity increased, the seeds expanded over the entire surface. In the *a*-IZO film produced by the solution process, 20 nm of the thin *a*-IZO layer was first formed from the bottom of the interface, regardless of the Zn molarity ratio. It is presumed that with an increase in the Zn concentration, an excess ZnO-aggregated layer sequentially emerges on the surface.

To determine the crystallinity characteristics, the XRD patterns of the fabricated *a*-IZO films were measured. Figure 3a shows representative peak analysis graphs of the IZO, InO, and ZnO films. The left-side graph of Figure 3a shows the measurement result in the normal mode (2θ-Ω mode), whereas the right-side graph shows the grazing incidence mode (GI mode). The main peak characteristics are shown in the graph. In this case, the measurement result in the normal mode indicates the crystallinity of the *a*-IZO/SiN_x_/Si bulk film, and the measurement result in the GI mode shows the surface crystallinity of the film. Here, the peak characteristic in the gray box of the left-side graph in Figure 3a is the lattice peak of the SiN_x_/Si bulk substrate; thus, it does not appear in the surface characteristic of the GI mode result of the right-side graph. Figure 3b shows the normal mode (bulk properties) and GI mode (surface properties) measurement results of the *a*-IZO thin film with respect to the Zn molar ratio. In addition to the Zn molarity characteristics of the *a*-IZO film, IZO and ZnO reference peaks are included at the top of each graph. It should be noted that unlike the results in Figure 3a, the bulk and surface characteristics of Figure 3b are completely different. These characteristics indicate that the crystallinity of the film has different structures in the bulk and at the surface. In the normal mode result of Figure 3b, similar crystal structures of the reference IZO film were obtained regardless of the Zn molarity ratio, whereas in the GI mode, the ZnO reference characteristic became prominent as the Zn molarity increased. It is speculated that for the *a*-IZO film produced by the solution process, a 20 nm of thin *a*-IZO layer is first formed at the bottom of the interface regardless of the Zn molarity ratio. Then, with an increase in the Zn concentration, excess ZnO aggregations are formed as multi-layers on the surface.

To determine the stoichiometric properties of the solution-processed *a*-IZO films, the XPS results were analytically measured. Figure 4 shows the XPS results, which were obtained by etching 10 nm from the top of the semiconductor surface. Figure 4a–c show the peak analysis of the bonding energy of Zn 2p, In 3d, and O 1s, respectively. Particularly in Figure 4c, the analysis peaks at 529.8, 531.0, and 532.0 eV are the main bonding energies, indicating the M-O lattice bonding, oxygen vacancy, and bonding energies of M-OH, respectively [29]. From the XPS analysis, the characteristics of Figure 4a,c showed a similar trend depending on the Zn concentration. In the 0.2 M sample, the Zn 2p peak was only slightly low, and the peak of the M-OH characteristic in the O 1s peak was relatively high. Considering the etched thickness of 10 nm, it can be inferred that only the 0.2 M sample shows the bonding-energy characteristics of the *a*-IZO layer, whereas the 0.3, 0.4, and 0.5 M samples show the characteristics of the ZnO aggregation layer. Particularly, a noticeable difference in the XPS analysis appeared in Figure 4b in 3d as the Zn molarity increased. The decrease in the In composition with respect to the Zn molarity is due to the increase in the thickness of the ZnO aggregation layer from the surface, which can be explained by the fact that most In atoms exist at the semiconductor–dielectric film interface.

The structural characteristics of the solution-processed *a*-IZO films with respect to the Zn molar ratio can be summarized as follows: To fabricate the IZO film by a solution process, it was assumed that a relatively uniform *a*-IZO film was formed on the SiN_x_ dielectric, with a small In concentration; Zn molarity of 0.2 M is considered an *a*-IZO film. As the Zn molarity increased, the excess Zn accumulated as an aggregation layer on the *a*-IZO film.

### 3.2. Growth of a-IZO Film

In addition to the structural characteristics of the *a*-IZO film, a chemical understanding of the reaction pathway of the solution process can help improve the physical properties (such as crystallinity, density, grain size, and composition) of the film. Figure 5 shows the AES measurement results from the surface of the film to 0–80 nm, measured for an *a*-IZO sample with a Zn molarity of 0.25 M. The dashed line in the graph represents the measurement result after spin coating (after 100 °C soft baking, which is the gel state), and the solid line represents the measurement result after annealing at 600 °C for more than one hour. The grey box in Figure 5 shows the boundary of the dielectric layer. In the AES results of the Zn and In atom concentrations, the structural characteristics of the multi-layered *a*-IZO film were clearly observed in the solid line. As shown by the solid line in Figure 5, 50% of the O atoms were uniformly observed in the film, and the Zn atom concentration was mostly observed on the upper surface of the film. The In atom concentration was mostly observed at the bottom semiconductor–dielectric film interface; it seems the irregular fluctuations at the surface are the effects of chemical contamination and morphology. The important growth characteristic of the annealing process from the *a*-IZO sol–gel state can be found in the characteristics of the rearrangement of atoms. In the dashed line, the atoms are relatively uniform in the gel state (after spin coating); however, the distribution of the Zn and In atoms is abruptly rearranged toward the top and bottom after annealing. This atomic rearrangement can occur because of the various reactions in the solution process, and in this study, a chemical reaction pathway of the nitrate precursor was applied for a more valid evaluation.

The stoichiometric analysis of the growth pathways of the solution-processed *a*-IZO film can be determined based on the decomposition process of the TGA graph. Figure 6 shows the TGA graph of the IZO solution used for the fabricated *a*-IZO film. The analysis of the nitrate-based thermal decomposition was referenced to the theory in this study [30]. As shown in Figure 6a, the metal–nitrate precursor with relatively high polarity loses chemical bonding in solution and is ionized according to Equation (1).
(1)$$M{\left({\mathrm{NO}}_{3}\right)}_{2}\cdot 3H_{2}O\to M{\left(\mathrm{OH}\right)}_{2}+2{\mathrm{NO}}_{3}^{-}+H_{2}O+2H^{+}$$

Moreover, in solution, nitrate forms a nitric acid–water azetrope via a hydrolysis/condensation reaction and can be expressed as Equation (2):(2)$${\mathrm{NO}}_{3}^{-}+3H^{+}+2e^{-}\to {\mathrm{HNO}}_{2}+H_{2}O$$

After spin coating, as the annealing temperature increased, a metal–nitrate-hydroxide (M-NO_3_-OH) structure was generated in the solution by the olation reaction, as shown in Equation (3) below, and the nitric acid–water mixture was removed by evaporation with a solvent at a temperature of ~120 °C, as indicated in Figure 6a (i) [31].
(3)$$M{\left({\mathrm{NO}}_{3}\right)}_{2}\cdot 3H_{2}O\to M\left(\mathrm{OH}\right)\left({\mathrm{NO}}_{3}\right)\cdot H_{2}O+{\mathrm{HNO}}_{3}↑+H_{2}O↑$$

As the temperature increases, the metal–nitrate bond is decomposed by the thermal decomposition reaction in Equations (4) and (5), and the remaining nitrate is exhausted as gas.
(4)$$4\mathrm{Zn}\left(\mathrm{OH}\right)\left({\mathrm{NO}}_{3}\right)\cdot H_{2}O\to 4\mathrm{Zn}{\left(\mathrm{OH}\right)}_{2}+4{\mathrm{NO}}_{x}↑+\left(5-2x\right)O_{2}↑+2H_{2}O↑$$
(5)$$4\mathrm{In}{\left(\mathrm{OH}\right)}_{2}\left({\mathrm{NO}}_{3}\right)\cdot H_{2}O\to 4\mathrm{In}{\left(\mathrm{OH}\right)}_{3}+4{\mathrm{NO}}_{x}↑+\left(5-2x\right)O_{2}↑+2H_{2}O↑$$

In the thermal decomposition of Equations (4) and (5), the decomposition temperature of the metal ion was affected by the magnitude of the charge density (CD). This charge density is proportional to the charge of ions and inversely proportional to the radius of atoms and is expressed as Equation (6) [18]:(6)$$\mathrm{CD}\sim \frac{\alpha }{T_{d}},\;\mathrm{CD}=\frac{3z}{4{\pi r}^{2}}$$
where α is a constant, $T_{d}$ is the decomposition temperature, and z and r are the atomic charge and radius (nm), respectively. According to Equation (6), the CD of In is ~1.41, and the CD of Zn is ~1.18. In Equations (4) and (5), the thermal decomposition of In, which has a relatively high CD, was observed at a lower temperature of ~184 °C (ii), and the thermal decomposition of Zn was approximately 233 °C (iii). It should be noted that nitrate has various by-products in solution and is removed as NO, NO_2_, and N_2_O gases. Depending on the final by-product of nitrate (NO_2_, N_2_O, O_2_, etc.), the thermal decomposition of nitrate can be completed at a relatively high temperature of ~335 °C (iv). Finally, an M-O-M oxo link structure (oxo linkage) is formed by the oxolation reaction in Equation (7):(7)$$M{\left(\mathrm{OH}\right)}_{2}\to \mathrm{MO}+H_{2}O↑$$

After the completion of reactions (ii) to (iv) in Figure 6b related with Equations (4) and (5), the oxidation reaction of Equation (7) sequentially occurs at a high temperature of 200–550 °C, and the condensation process of the film proceeds. To form an improved M-O-M oxo linkage through oxidation, annealing was carried out up to 600 °C. Consequently, it can be presumed that the cause of the multi-layer *a*-IZO/ZnO film, as discussed in the structural characteristics, is closely related to the subsequent occurrence of the thermal decomposition of the In-nitrate and Zn-nitrate at different temperatures.

The growth characteristics of solution-processed *a*-IZO films can be summarized as follows. The sol film formed by spin-coating was sintered into a gel film through solvent and azetrope evaporation. In gel-type *a*-IZO thin films, nitrate is terminated as a gas through thermal decomposition, and a multi-layer film is formed in the order of the *a*-IZO semiconductor and ZnO aggregation layers. Subsequently, the remaining -OH is removed in the form of H_2_O and O_2_ through oxidation, and the thin film is finally condensed in the form of an M-O-M oxo-link structure.

### 3.3. Electrical Behavior

The electrical performance of the *a*-IZO TFT was evaluated with respect to the Zn molar ratio. Figure 7 shows the (a) transfer characteristics, (b) output current, and (c) electrical parameters of the fabricated *a*-IZO TFT with respect to the Zn molarity ratio. As shown in Figure 7, the on-state current of the transfer and output characteristics gradually decreased as the Zn molarity increased, and the threshold voltage monotonously increased in proportion to the log-scale molarity. In TFTs with a high Zn molarity of 0.5 M or higher, it was observed that the influence of the gate voltage was reduced, so that the on-state current control did not properly work. The maximum value of the field-effect mobility of TFT and the maximum value of the on/off ration were observed in the TFT with a Zn molarity of 0.3 M. Additionally, the devices with a low Zn molarity below 0.3 M showed rather high off-state current. The high off-state current of the TFT 0.2–0.25 M is attributed to the high gate leakage current from the junction [32].

Eventually, the decrease in the electrical performance of the TFT due to the increase in Zn molarity can be interpreted as an increase in the total series resistance owing to the ZnO aggregation layer, as shown in Figure 8. Figure 8 shows a schematic of the atomic structure and growth process of the solution-processed *a*-IZO thin film. Figure 8a shows the ionic states that may exist in the gel state after spin coating and the simplest atomic bonding model by decomposition. Additionally, (b) shows the multi-layer structure of the *a*-IZO film and the atomic bonding model of the by-product after annealing. In Figure 8a, the number of covalent bonds in the amorphous IZO material is determined by the bonding coordinate, and because of defects in the amorphous random network, there can be imperfect bonds, such as weak bonds, dangling bonds (D_InO_), and oxygen vacancies (V_O_) [33,34]. Particularly for the In-O bond, as shown in the simplest atomic model in (a), a three-fold coordinated In atom replaces the Zn atoms. Thus, an In^+^–D_InO_^−^ bond can be formed, which can act as a donor via D_InO_^–^→D_InO_^0^ + e^−^.

Based on the structural characteristics, chemical growth, and electrical characteristics of the *a*-IZO film with respect to the Zn molarity ratio, the cause of the multi-layer thin-film structure in Figure 8b can be inferred as follows. First, while annealing the IZO gel, the In-O oxo link was organized at a relatively low temperature (T_d_In_ ~183 °C) by thermal decomposition. Simultaneously, ~10^11^ cm^−3^ of N^+^ fixed charges in the dielectric and D_InO_^−^ charges are attracted to each other by Coulombic interaction, and an In-N covalent bond is formed through the annealing process. The extra electrons increased the free electron density in the *a*-IZO film. Through this process, it can be speculated that ~20 nm of the *a*-IZO film was formed at the semiconductor–dielectric interface. Subsequently, at a relatively high thermal decomposition temperature (T_d_zn_ ~237 °C), the ZnO amorphous network structure started to emerge as a ZnO aggregation layer. Consequently, an *a*-IZO semiconductor layer, which determines the electrical performance, was initially formed regardless of the Zn molar ratio. The excess ZnO aggregation became thicker with respect to the Zn molar ratio. Consequently, this ZnO aggregation layer acts as a series resistance R_ZnO_, which prevents charge injection into the TFT channel. Additionally, the effective voltage between the gate and drain sources is reduced, and the threshold voltage is increased because of the effects of the series resistance. The decrease in the electrical performance of the TFT in the transfer curve and output current can be explained by the effect of the ZnO aggregation layer, as discussed.

## 4. Conclusions

In this paper, we highlighted the relationship between the structural characteristics and electrical behavior of solution-processed *a*-IZO TFT with respect to the Zn molarity, and a chemical growth pathway of the *a*-IZO film was proposed. In the *a*-IZO film fabricated using the solution process, an *a*-IZO layer with a thickness of approximately 20 nm was initially formed regardless of the Zn molarity ratio. As the Zn concentration increased, the excess ZnO aggregated on the surface of the film as a multi-layer structure. The structural characteristics of the *a*-IZO film were confirmed through the measurement results of SEM, XRD, and XPS. Furthermore, the solution-processed *a*-IZO film was in a gel state uniformly dispersed after spin coating and was grown into a multi-layer having a heterogeneous structure through the annealing process. The characteristics of chemical growth of the *a*-IZO film were analyzed through the data of AES and TGA. The multi-layer structure was grown from a uniform gel state through annealing, and the properties of the In and Zn element rearrangements were stoichiometrically defined through a chemical reaction pathway. It is speculated that the multi-layer structure is originated from the different decomposition temperatures of each In and Zn material, and coulombic interaction between the fixed charge and dangling bond of the dielectric and In-O bonding structure. Finally, it was inferred that the lowered electrical performance of the *a*-IZO TFT depending on the Zn molarity was because the ZnO aggregation acted as a series resistance. Further detailed research on the effects of ZnO structure on the charge transport behavior in an *a*-IZO semiconductor can be developed by excluding the influence of film thickness itself on the threshold voltage and on-state current of TFTs. We believe that the structural and electrical characterizations of nitrate-based solution-processed AOS in this study can be applied to further improve the feature characteristics of solution-processed oxide semiconductors having multi-layer structures.

## Figures and Tables

**Figure 1 materials-15-03416-f001:**
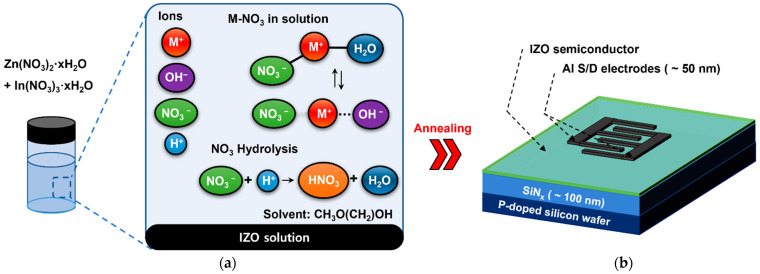
(**a**) IZO solution based on the metal–nitrate precursor used for manufacturing *a*-IZO TFT, and the types of ions and by-products that can be produced by dissolution. (**b**) Schematic illustration of finger-type solution process *a*-IZO TFT.

**Figure 2 materials-15-03416-f002:**
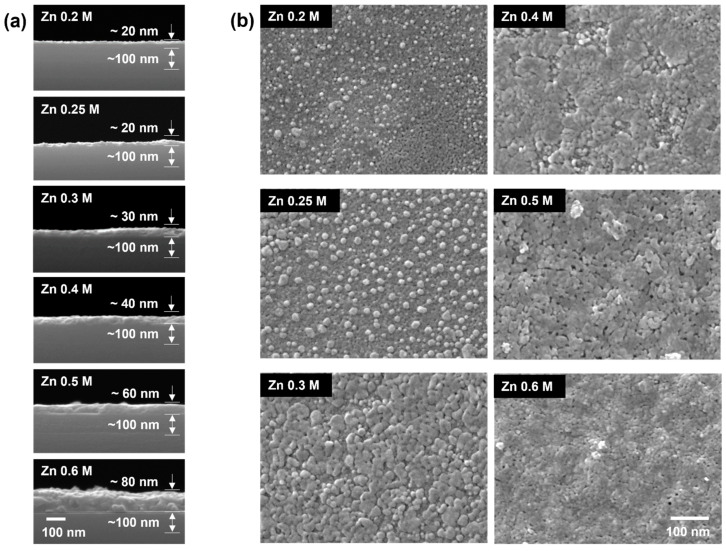
(**a**) Cross-section and (**b**) morphological SEM images of *a*-IZO films with respect to the Zn molarity ratio. The In molarity ratio used for the fabrication of each *a*-IZO film was 0.05 M.

**Figure 3 materials-15-03416-f003:**
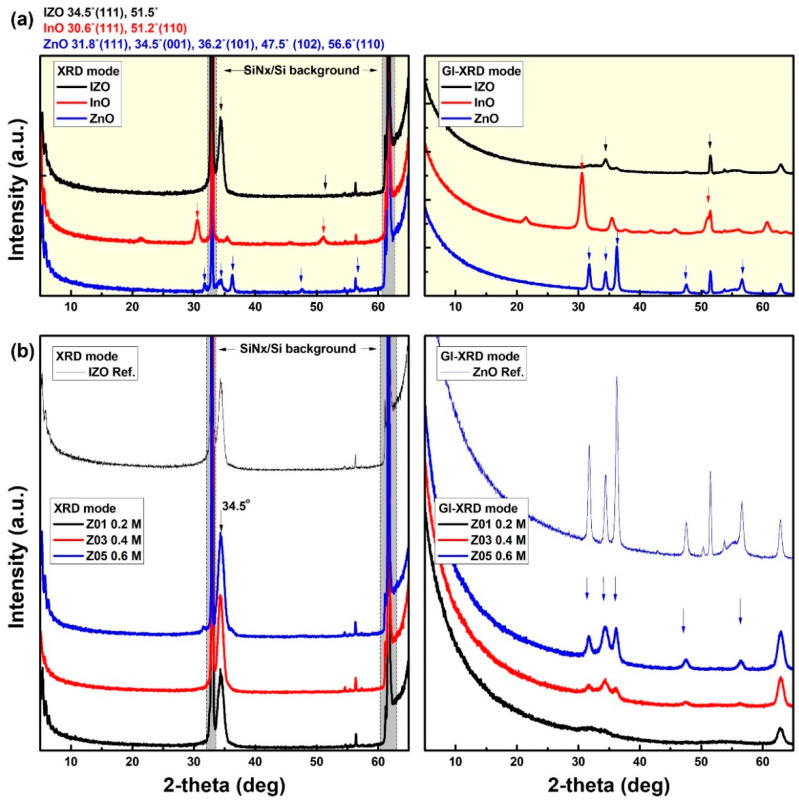
XRD graph to analyze the crystallinity of *a*-IZO films. (**a**) Reference peak analysis of IZO, InO and ZnO films. (**left**) Crystallinity of bulk film measured in 2-θ mode. (**right**) Surface crystallinity measured in grazing incident (GI) mode. Several reference peaks of IZO, InO, ZnO films are indicated at the top of the graph (**a**). (**b**) (**left**) Bulk and (**right**) crystallinity of the *a*-IZO films with respect to Zn molarity.

**Figure 4 materials-15-03416-f004:**
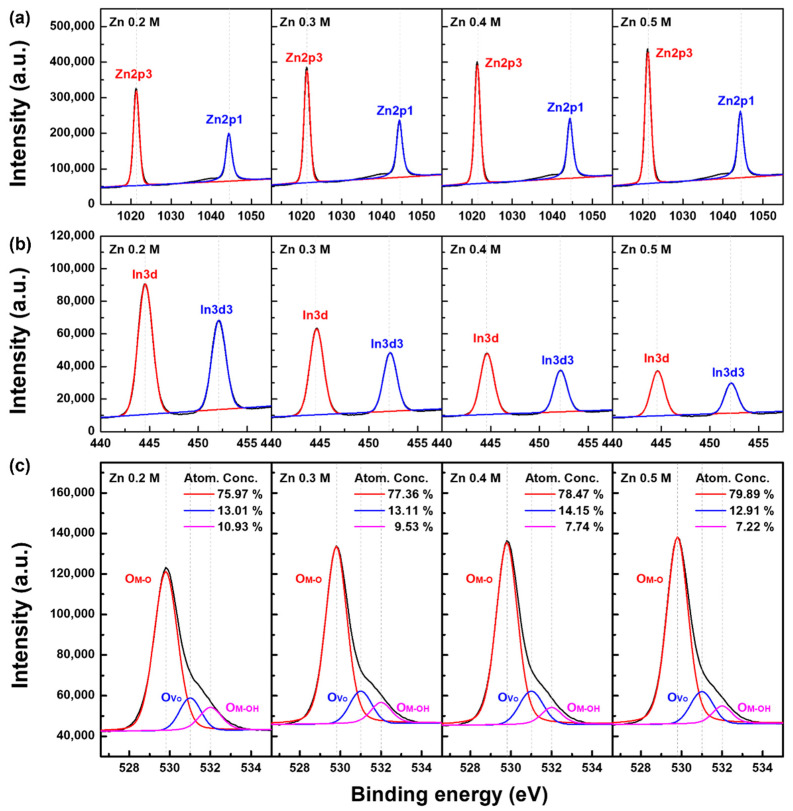
XPS results for stoichiometric characterization with respect to the Zn molarity. (**a**) Zn 2p, (**b**) In 3d, and (**c**) O 1s peak analysis of *a*-IZO films. The bonding energies of 529.8, 531.0, and 532.0 eV suggest M-O, oxygen vacancy (V_O_), and M-OH bond, respectively.

**Figure 5 materials-15-03416-f005:**
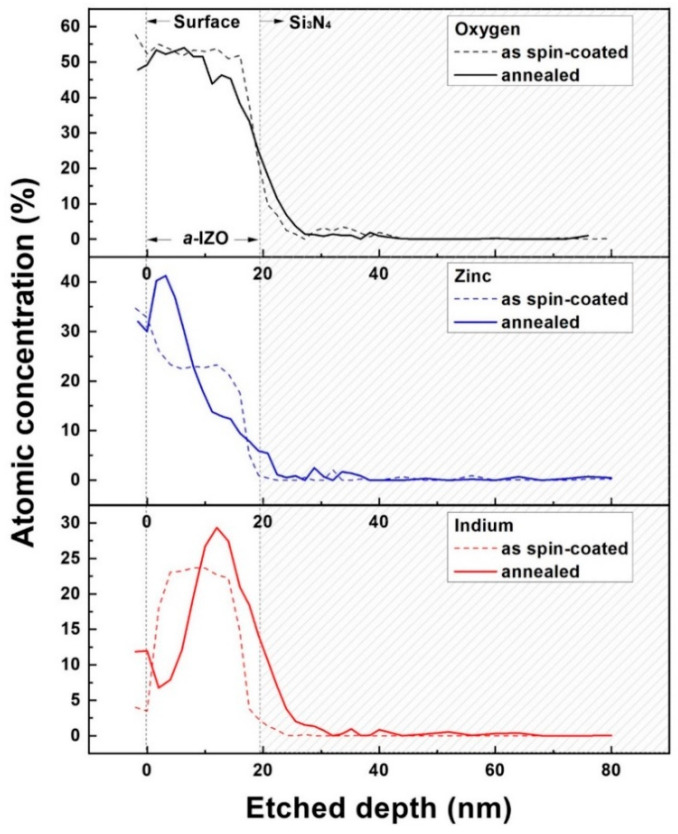
AES graph in terms of the depth of *a*-IZO film. Measured results (dashed line) after spin coating and soft baking (110 °C) and (solid line) after annealing at 550 °C.

**Figure 6 materials-15-03416-f006:**
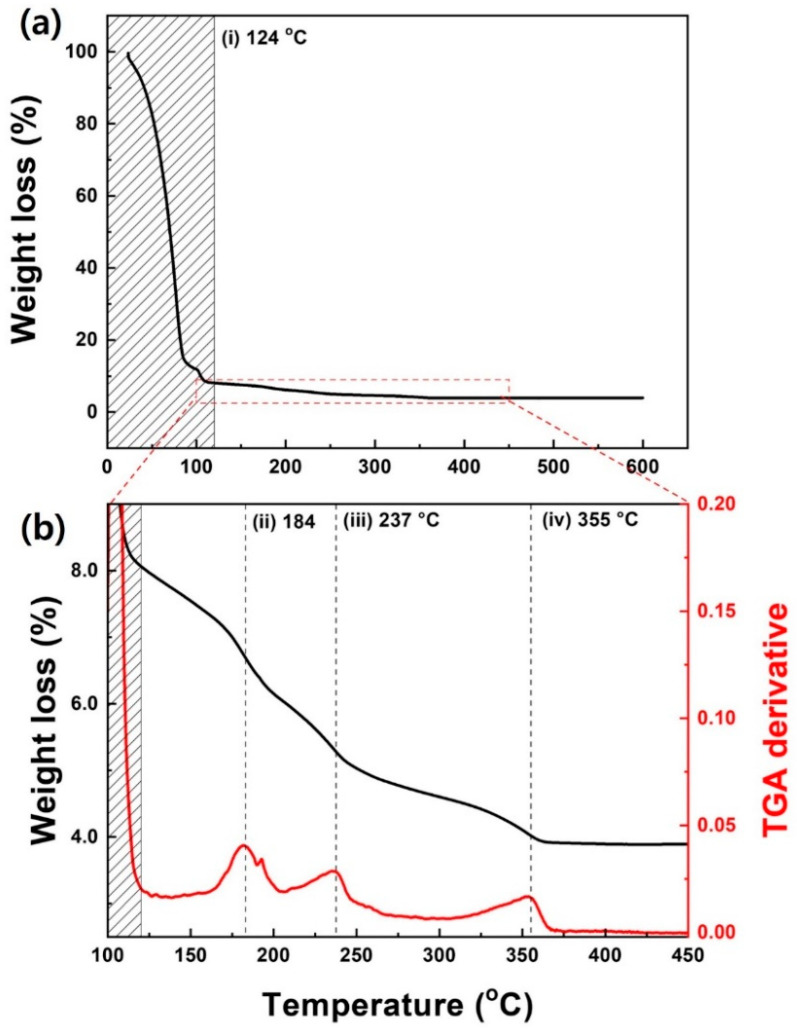
TGA result of IZO solution for *a*-IZO film fabrication. (**a**) Weight loss versus temperature graph, and (**b**) enlarged weight loss/differential TGA graph at 100–450 °C.

**Figure 7 materials-15-03416-f007:**
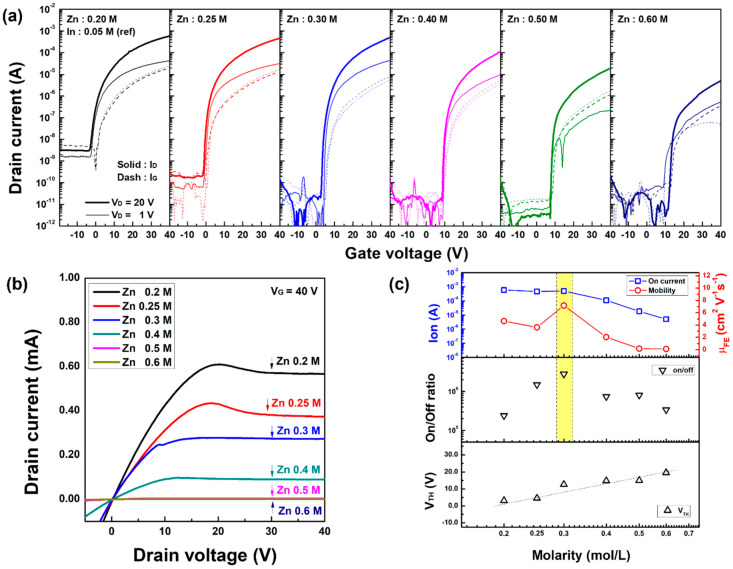
Electrical characteristics of solution-processed *a*-IZO TFT with respect to the Zn molarity. (**a**) Transfer curves, (**b**) output currents, and (**c**) electrical parameters of the TFT. In (**a**), the thick/thin lines represent saturation/linear conditions and solid/dashed lines indicate I_D_/I_G_, respectively.

**Figure 8 materials-15-03416-f008:**
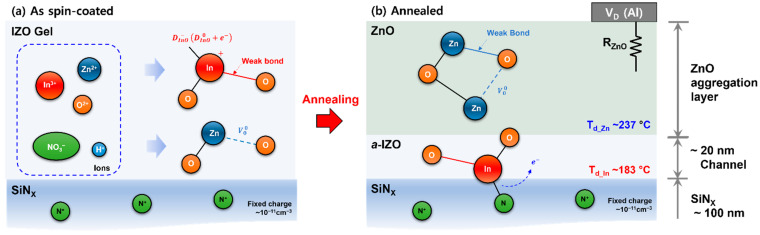
(**a**) Illustration of types of ions in gel-state *a*-IZO film after spin coating and the simplest atomic structure by oxidation. (**b**) Schematic of multi-layer *a*-IZO film and possible atomic structures of by-product through the annealing process.

**Table 1 materials-15-03416-t001:** Molarity table of IZO solution for *a*-IZO TFT fabrication (the In molarity of solution is fixed).

No.	Zn01	Zn02	Zn03	Zn04	Zn05	Zn06	Zn07	Zn08
Zn molarity (M)	0.1 (failed)	0.15 (failed)	0.2	0.25	0.3	0.4	0.5	0.6
In molarity (M)	0.05 M (fixed)							

## Data Availability

The research data presented in this study are available on request from the corresponding author.

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
