# Peer review of "Atomic Structure Evaluation of Solution-Processed a-IZO Films and Electrical Behavior of a-IZO TFTs"

_materials, 2022, doi:10.3390/ma15103416_

Round 1

Reviewer 1 Report

This paper verifies the conclusion that “regardless of the Zn molarity ratio, the first 20-nm-thick a-IZO film is formed in the bottom layer, and then excess ZnO aggregates in the top layer as the Zn concentration increases” by various test methods. However, for the discussion that the ZnO aggregation layer increases the total series resistance in equivalent circuit and affects the electrical properties, the film thickness needs to be excluded as a factor so that it would be more convincing. It is suggested to add a set of control experiment by fabricating devices with film thicknesses of 30/40/60/80 nm using Zn01 solution (Zn 0.1M, In 0.05M) to exclude the effect of film thickness itself on the threshold voltage and on-state current.

Author Response

Thank you very much for your valuable comments.

We reflected your comments in the revised manuscript.

Best Wishes

Reviewer 2 Report

Ref.comments to the paper titled as “Atomic Structure Evaluation of Solution-Processed a-IZO Films  and Electrical Behavior of a-IZO TFTs” written by the authors: Dongwook Kim, Hyeonju Lee, Bokyung Kim, Jin-Hyuk Bae, Jong Sun Choi, and Sungkeun Baang

It is known that the properties of the composite components of transistors depend both on their individual features and on the superposition of properties, which can improve or level the basic technical parameters of the subsequent device. Thus, it is really true that the understanding the chemical reaction pathway of the metal-salt precursor is essential for the modifying the properties of the solution-processed metal-oxide thin films and further improving their electrical performance. From this point of view the manuscript is actual and modern.

For the first, it is remarked that the author has made the literature search, analyzing 22 references. Indeed, this can indicates the knowledge of the problem, its useful application and finding the different ways to solve it. But, it is not enough for this perspective material based on a-IZO films. The analysis of the papers (5-7 publication as an additional) written last 3-5 years should be added.

Methods and experiment parts are good. They are good illustrated, and supported with pictures, table data, and the large numbers of the devices to study the technical parameter of the materials.

In Results and Discussion section it is interesting and useful. SEM images of a-IZO films are shown; XRD analysis are supported the crystallinity of a-IZO films; AES curves made in the in terms of the depth of a-IZO film are shown in order to present the experimental results after spin coating technique, soft baking (at 110°C) and after annealing at 550°C; TGA result of IZO solution for a-IZO film fabrication are not contradicted with our basic physical knowledge.  Electrical characteristics of solution-processed a-IZO TFT with respect to the Zn molarity are very interesting not only for the engineering, but for the education process as well. In this concern, the charge transfer mobility can be measured or calculated via, for example, via the Childe-Lengmuir law (please see F. Gutman, L.E. Lyons, Organic Semiconductors (Wiley, New York, 1967).

Conclusion part accumulates some data, but it is not enough. Conclusion should be extended.

So, the paper is interesting for the specific area for the researchers and students. I can recommend to the authors to answer the questions mentioned above. Thus, the paper can be published after minor corrections.

Author Response

(The authors gave the same response as above.)
